# Epidemic dynamics of respiratory syncytial virus in current and future climates

Rachel E. Baker [1]*, Ayesha S. Mahmud[2,8], Caroline E. Wagner[3], Wenchang Yang [4], Virginia E. Pitzer[5], Cecile Viboud[6], Gabriel A. Vecchi [1,4], C. Jessica E. Metcalf [3,7] & Bryan T. Grenfell[3,6,7]

A key question for infectious disease dynamics is the impact of the climate on future burden. Here, we evaluate the climate drivers of respiratory syncytial virus (RSV), an important determinant of disease in young children. We combine a dataset of county-level observations from the US with state-level observations from Mexico, spanning much of the global range of climatological conditions. Using a combination of nonlinear epidemic models with statistical techniques, we find consistent patterns of climate drivers at a continental scale explaining latitudinal differences in the dynamics and timing of local epidemics. Strikingly, estimated effects of precipitation and humidity on transmission mirror prior results for influenza. We couple our model with projections for future climate, to show that temperature-driven increases to humidity may lead to a northward shift in the dynamic patterns observed and that the likelihood of severe outbreaks of RSV hinges on projections for extreme rainfall.

[1] Princeton Environmental Institute, Princeton University, Princeton, NJ, USA. [2] Planetary Health Alliance, Harvard University, Cambridge, MA, USA.
[3] Department of Ecology and Evolutionary Biology, Princeton University, Princeton, NJ, USA. [4] Department of Geosciences, Princeton University, Princeton, NJ, USA. [5] Department of Epidemiology of Microbial Diseases, Yale School of Public Health, New Haven, CT, USA. [6] Division of International Epidemiology and Population Studies, Fogarty International Center, National Institutes of Health, Bethesda, MD, USA. [7] Woodrow Wilson School of Public and International Affairs, Princeton University, Princeton, NJ, USA. [8] Present address: Department of Demography, University of California, Berkeley, Berkeley, CA, USA.
*email: racheleb@princeton.edu

Understanding the climate drivers of infectious diseases has clear public health implications, often delineating our ability to predict the timing, location and magnitude of outbreaks both currently and in the future as the climate changes[1]. Important steps have been taken to decompose the climatic drivers of vector-borne[2] and waterborne diseases[3,4], and to develop projections for the future burden and geographical range of pathogens across climate change scenarios[5]. Less attention has been paid to directly-transmitted infections, despite key work on influenza demonstrating the significant role climatic variables can play in driving the transmission of these types of infections. Experimental studies show that influenza transmission increases in conditions of low humidity[6–8]. Observational studies in temperate climates support this finding; while in tropical locations, influenza peaks have been observed to occur in the rainy season when humidity increases[9,10]. The rapid evolution of the seasonal influenza virus and the consequent complexity of dynamics makes projections of future epidemics difficult[11,12]. In contrast, many major childhood infections exhibit more regular dynamics, making the long-term consequences of environmental drivers easier to dissect[13–16].

Respiratory syncytial virus infects most children before the age of two and is a leading cause of lower respiratory tract infections in infants, particularly bronchiolitis and pneumonia, and has also been implicated in the later life development of asthma[17,18]. Recent estimates suggest that RSV is responsible for ~5.2% of under-five deaths globally[19], though the proportion may be even higher for younger age groups experiencing their first infection[20,21]. RSV epidemics exhibit distinct seasonal cycles[22], with the peak occurring around the winter months in temperate locations and closer to the rainy season in tropical, more humid locations[23–25], as has been noted for influenza[9,26]. Recent work by Pitzer et al., using state-level data from the United States, showed broad-scale correlations between several annually-averaged climate variables and the amplitude of the periodic forcing of the modeled RSV cycle, suggesting that climate may play a crucial role in driving epidemics[27]. However, the results did not address the distinct seasonality of RSV in tropical locations, nor the earlier timing of epidemic onset in Florida. Higher resolution data over a wider spatial scale is required to disentangle the specific climatic drivers governing the dynamics and timing of RSV epidemics and to unite predictions for both temperate and tropical locations.

Here, we leverage a county-level dataset of weekly RSV hospitalizations from over 300 locations in the United States and combine these observations with weekly bronchiolitis hospitalization data from all 32 states in Mexico. The temporal range of our dataset spans 5–15 years depending on location (Fig. 1a). The spatial extent of our dataset covers a diverse set of climatologies including tropical, humid locations in southern Mexico and temperate climates in the US, where locations in northern latitudes experience large seasonal variation in climate conditions (Supplementary Fig. 1 and 2). We combine our RSV case data with high resolution climate data (precipitation, specific humidity, and temperature) in order to investigate spatial patterns in dynamics, evaluate the drivers of transmission, and project future RSV cycles under climate change scenarios.

## Results

**Current climate.** Spatial order is observed in the timing of onset of the RSV epidemic and the dynamic patterns of incidence (Fig. 1b, e). Mid-western and northern counties in the United States have biennial cycles which alternate between high and low incidence years (Fig. 1c)[27]. Coastal and southern US locations exhibit annual cycles of incidence where cases peak seasonally

and then disappear in the off-season. Counties in Florida and most Mexican states also experience annual cycles of incidence, although the epidemic in these locations persists throughout the year resulting in a shallow-trough cycle. The earliest onset of the epidemic occurs in the summer months simultaneously in Miami, Florida, and the Yucatan peninsula in Mexico. Onset timing then follows a latitudinal gradient northward such that onset occurs last in the northern US in late December (Fig. 1b, Supplementary Fig. 5). The timing of school semesters is unlikely to drive this gradient: while US school start dates vary by state, in Mexico the first day of the school year is more broadly standardized across the country[28].

We find that timing of onset is correlated with mean local climate (Fig. 1d). There is no animal model for RSV so we rely on mechanistic understanding developed from influenza experiments to inform our choice of climate variables (Supplementary Note 1). Therefore we begin our analysis by considering specific humidity and include precipitation because of its observed importance in tropical locations[9,23–26]. We also test for effects of temperature as a fundamental meteorological variable (though it is functionally related to specific humidity: Supplementary Note 1). Specific humidity, precipitation, and temperature are significantly ($p \ll 0.001$) associated with the mean timing of onset of the RSV epidemic, with specific humidity explaining 52% of the variance in mean onset timing across locations. Southern Florida and the Yucatan peninsula, where the outbreak starts, are the most humid regions in the dataset. To investigate whether year-to-year variations in humidity can alter the timing of onset of the epidemic within a particular location, we calculate the onset week for every year for each location in our dataset, and fit a regression model including dummies for each location to remove mean onset timing. We find that a 1 (g/kg) increase in mean annual specific humidity shifts the timing of the RSV epidemic back by 1 week ($p \ll 0.001$). Similar effects are shown for both temperature (0.3 week/°C) and average precipitation (0.5 week/mm) (Supplementary Table 1).

While these spatial level correlations are striking, it is necessary to characterize the climate drivers of transmission in order to model possible future changes to RSV dynamics. We calculate an empirical transmission rate based on the discrete time version of the susceptible–infected–recovered (TSIR) model using $\mathrm{Em}\beta_t = \frac{I_{t+1}N_t}{I_t^\alpha S_t}$, assuming we can approximate the first RSV infection as immunizing given that secondary infections are less severe and less infectious[29,30]. Here we observe $I_t$, $I_{t+1}$, and $N_t$ directly from our data where $I$ is incidence, $N$ is population and $t$ is the generation time of RSV, approximated as 1 week[27]. $\alpha$ captures heterogeneities in mixing and the effects of discretization. In order to estimate $S$, the number of susceptibles, which is not directly observed within our dataset, we fit the TSIR model to each location and extract the time series of susceptibles from the mean model fit.

The constructed dataset of empirical transmission rates is included in a panel regression model, where log transmission is the dependent variable and climate variables are the independent variables. Our main regression specification includes a suite of controls: specifically, county-by-month dummies to remove potential confounding from location-specific seasonal aggregation such as school semesters, and county-by-year dummies to remove confounding from location-specific episodic events. As in the descriptive analysis, we include both precipitation and specific humidity in our baseline regression model. Results for temperature are included in Supplementary Fig. 10 and mirror an expected logarithmic functional transformation of the humidity result. We find a significant effect ($p \ll 0.001$) of both precipitation and specific humidity on RSV transmission (Fig. 2a, Supplementary

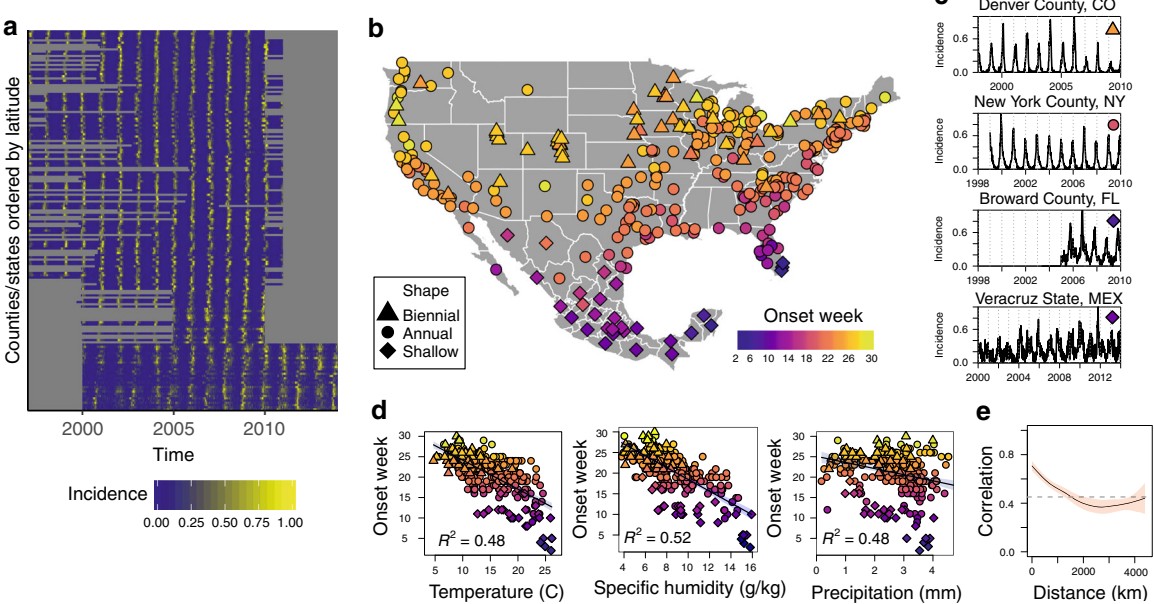

**Fig. 1** Broad-scale patterns of RSV are correlated with local climate. **a** Incidence time series for each county (USA) and state (Mexico) in the dataset. **b** Timing of onset (color) and dynamic pattern (shape) of each location in the dataset. **c** Example incidence time series for four location exhibiting distinct dynamic patterns. **d** Correlation between timing of epidemic onset and mean climate conditions (averages over temporal range of data). **e** Spatial correlation between RSV time series.

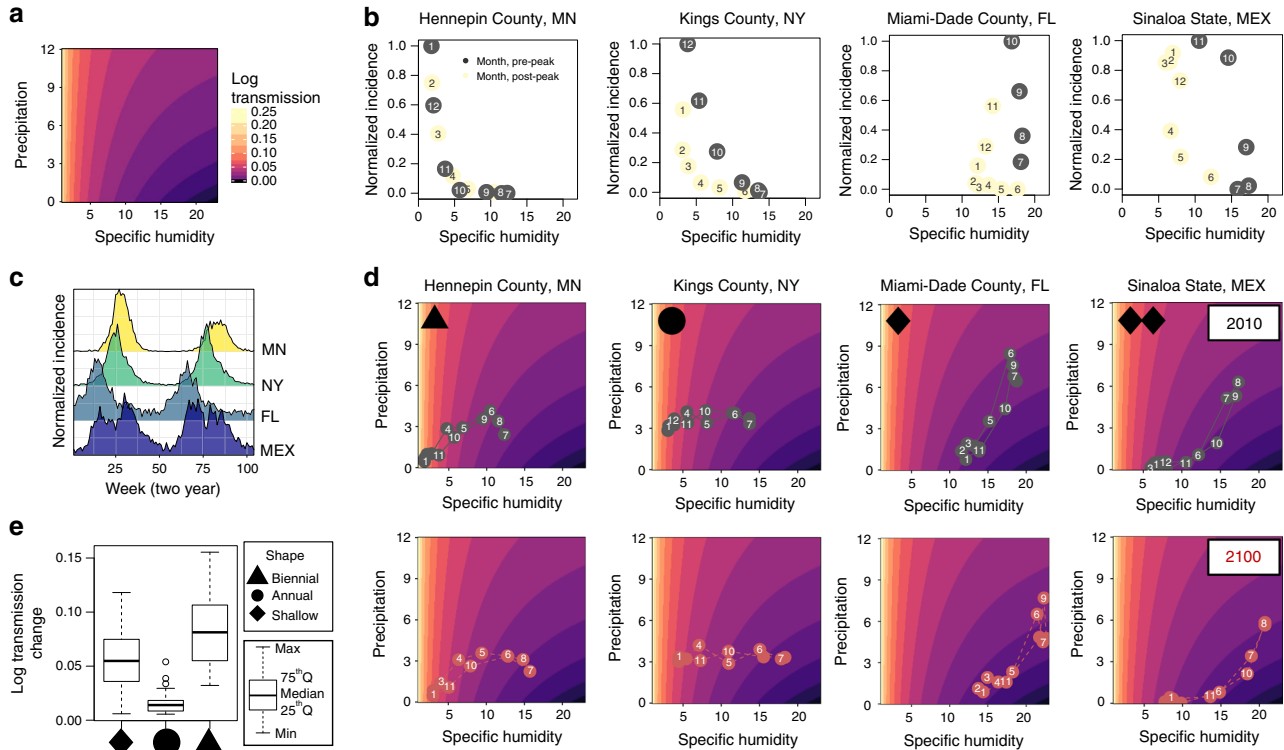

**Fig. 2** Specific humidity and precipitation drive RSV transmission. **a** Model results for the predicted effect of specific humidity and precipitation on transmission. **b** Normalized monthly RSV cases before peak incidence (gray) and after peak incidence (yellow) against mean monthly humidity for four locations in the dataset, month-of-year shown in point. **c** Mean incidence averaged biennially for the same four locations demonstrating distinct dynamic patterns. **d** Seasonal trajectory in terms of humidity and precipitation for same locations in 2010 and 2100, month-of-year shown in point. **e** Boxplot showing seasonal change in transmission grouped by dynamic pattern.

Table 2). Fitting a flexible nonparametric model reveals that humidity follows an inverse relationship with log transmission (Supplementary Fig. 8 and 9). Peak transmission occurs in very dry conditions and drops sharply as humidity increases. In contrast, we find a positive linear effect of rainfall on transmission. Our combined results imply that the humidity effect will dominate in locations that experience very low humidity at certain times of year, while rainfall will become a key driver in locations where humidity is on average higher or varies minimally over the year, such as in tropical climates.

We consider the implications of our estimated climate effect for four locations in our dataset with exemplar dynamic patterns and timing of epidemic onset (Fig. 2b, c). In Fig. 2d we show the seasonal trajectories for these locations, with the mean monthly rainfall and humidity overlaid on our predicted transmission effect. Hennepin County, MN, an example of a location with biennial outbreaks, has large changes in predicted transmission over the course of the year, with maximum transmission in the winter months (Fig. 2c) when peak incidence also occurs (Fig. 2b). Kings County, NY, an example of a location with annual outbreaks, has a wintertime peak in predicted transmission driven by declining humidity, though not to the same extent as the biennial test case. In contrast, our model predicts that Miami-Dade County, FL, an example of a shallow-trough location, experiences peak transmission in the summer months driven by summertime rainfall. The dominance of the rainfall effect means that Miami experiences an increase in cases even as humidity increases, a reversal of the patterns observed over most US counties (Fig. 2b), which explains the early onset of the epidemic in this location.

Many Mexican states show a similar result to Miami. However, for some states such as Sinaloa our model predicts two optimal times for transmission within a year, one driven by low humidity and the other driven by increasing rainfall. These two peaks are visible in the Sinaloa incidence time series (Fig. 1c). While such patterns have been previously observed in both RSV and influenza time series in tropical countries[9,23], our results provide evidence for the climatic variation driving these observed dynamics. Across the dataset we find that locations with biennial dynamics experienced larger seasonal changes in predicted transmission than locations with shallow-trough dynamics (Fig. 2e) as suggested by earlier work[27].

**Future climate.** In our historical analysis we made the conservative assumption of removing local seasonal variation and local year-to-year mean variation with the inclusion of dummy variables, essentially looking at the effect of climate anomalies on transmission. These dummy variables subsume both annual climate variations and social drivers of transmission. In future projections we refit the model without these dummy variables to explore the maximum likely effect of climate on future transmission (Supplementary Fig. 12, using a common seasonal effect across locations to capture schooling generates similar results: Supplementary Fig. 14). We run simulations varying the mean transmission rate and the amplitude of the seasonal change in transmission, holding birth rates and population constant (Fig. 3a left). As the amplitude increases, cycles tend to become more biennial and eventually chaotic[31,32]. Figure 3a (right) shows the trajectory of each county in terms of current to future predicted cycle dynamics, where the future is the end-of-century Coupled Model Intercomparison Project Phase 5 (CMIP5) multi-model-mean climate, based on the Representative Concentration Pathway 8.5 (RCP8.5) scenario[33]. We find that increases to specific humidity, driven by higher temperatures, result in reduced transmission and a lower amplitude of seasonal transmission,

with locations in our dataset no longer experiencing the very dry conditions that drive larger seasonal transmission changes. This results in cycles becoming more annual and shallower on average: a northward latitudinal shift in the types of dynamic patterns observed.

While individual CMIP5 models are consistent in projecting increasing surface air temperature and hence specific humidity, there is significant disagreement across models with regard to future changes to regional precipitation[34]. Given that precipitation is found to be an important driver of RSV, particularly in tropical locations, we explore the implications of the model range in precipitation projections for future RSV epidemic dynamics. In Fig. 3b we show simulations of the RSV cycle for three locations in Mexico where transmission is driven by humidity, fixed at the 2100 CMIP5 multi-model-mean and precipitation, based on trajectories from all 23 CMIP5 models (three models did not include explicit precipitation projections and are removed). The upper 90th percentile, the 50th percentile, and the lower 10th percentile, in terms of the amplitude of RSV transmission predicted by the model, are highlighted within the figure.

The uncertainty in the size of simulated epidemics depends on the relative role of precipitation in driving the transmission cycle in a particular location. For instance, the seasonal cycle of RSV in Sonora is mostly driven by changes to humidity (Supplementary Fig. 11), meaning that deviations from mean precipitation do not alter cycle dynamics. However, in Oaxaca, where humidity remains constant throughout the year, differences in future rainfall across CMIP5 models result in divergent cycle dynamics. High precipitation results in a bifurcation, leading to biennial outbreak cycles.

We evaluate the CMIP5 model uncertainty by calculating differences between individual 2100 model projections and 2010 estimates of the seasonal change in transmission across all locations in the dataset (Fig. 3c). For many locations in the United States, model differences in rainfall projections do not alter the mean decline in seasonal transmission change. In these locations the RSV cycle is driven primarily by changes to humidity. However, for Mexico, Florida, and coastal parts of the western US, there is a large divergence across models and extreme rainfall increases the seasonal transmission change in multiple scenarios. In these locations, the future dynamics of RSV will hinge on future rainfall, and reducing uncertainty across climate models will help better predict and prepare for potential changes to RSV epidemic intensity.

## Discussion

Our model unites predictions for the climate drivers of RSV across both temperate and tropical locations, finding that the effect of low humidity trades off against rainfall drivers, depending on location. The striking similarity in the role of specific humidity for both influenza and RSV suggests that parallel mechanisms are at play for the two viruses (Fig. 4); however, the lack of an animal model for RSV means experimental work cannot currently be used to support our findings. This correspondence appears to hold even though the population dynamics of influenza and RSV are palpably different. Influenza disappears at high latitudes in the summer whereas RSV is much more endemic. Disentangling the roots of these differences is an important area for future work. Precipitation has also been hypothesized to be important for influenza transmission, although this has not been tested in laboratory settings, likely due to the relative complexity of experimental set-up[35]. It is unknown whether physical processes such as aerosol deposition, or behavioral processes such as grouping indoors, explain the mechanism of precipitation drivers of transmission for both RSV and influenza.

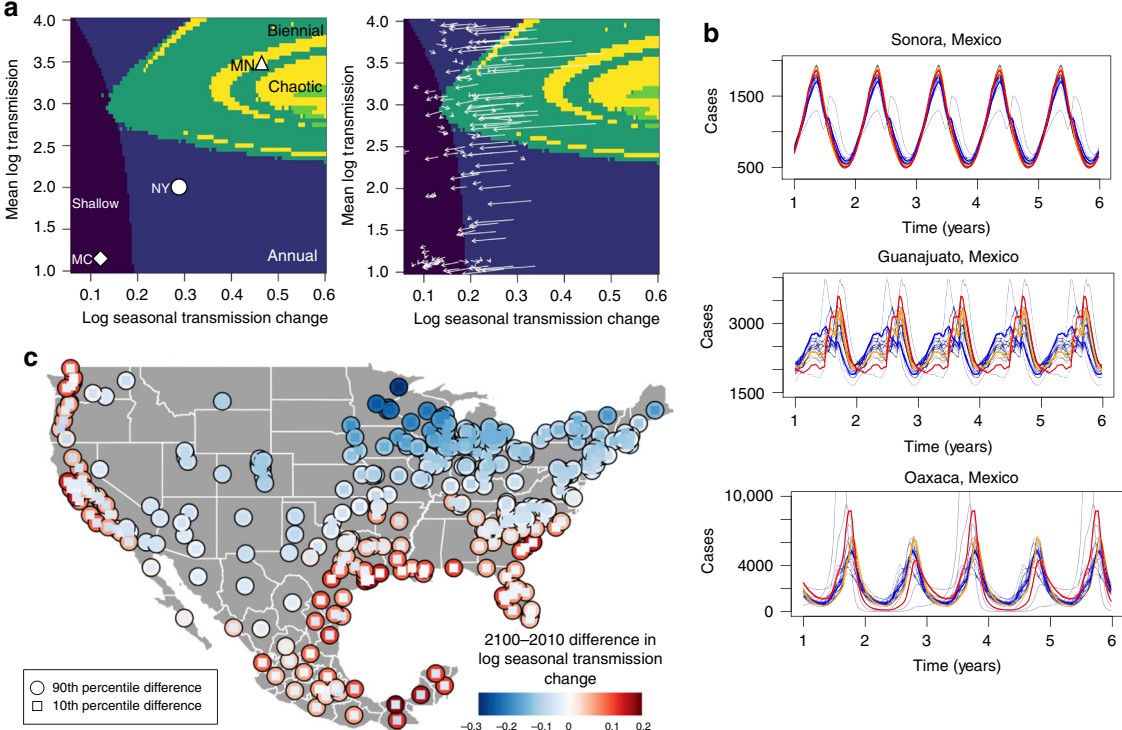

**Fig. 3** Projections of RSV dynamics under different climate scenarios. **a** Simulated effect of log mean transmission and log seasonal transmission change (annual max–min transmission values) on dynamic pattern, holding births and initial population constant, with trajectories of current and future climate shown with arrows on the right plot. **b** Simulated epidemic dynamics for three states in Mexico using precipitation projections from all climate models (gray lines). Ninetieth percentile (red), 50th percentile (orange), and 10th percentile (blue) in terms of the amplitude of seasonal transmission are highlighted. **c** Map of the uncertainty in the projected 2100–2010 difference to log seasonal transmission change where outer circles is the upper 90th percentile and inset blue squares are the 10th percentile projections.

There are important caveats to our results. We made the decision to remove dummy variables in future projections so that annual variation in forced climate could flexibly determine transmission. However, this means our simulations do not account for non-climatic factors that may also structure transmission. We crudely tested this by using a common seasonal fixed effect which gave qualitatively similar results, reinforcing the robustness of our assumption. Developing a richer understanding of the mechanisms connecting climate to transmission would help validate our results further. Our model also does not take into account genetic sub-type of RSV[30] due to data limitations in the US and Mexico. We also do not consider age structure or potential secondary infections: secondary RSV infections are less severe[27]. Nonetheless, we are able to capture the limit cycle structure of RSV dynamics well and elucidate multiple streams of evidence for a fundamental climate effect.

The public health implications of changes to future RSV epidemic dynamics will depend on location-specific climatic changes. Precipitation-driven increases to epidemic intensity, caused by climate change or other large-scale climatic events such as the El-Nino-Southern Oscillation, will require increased surge capacity in locations that typically observe more uniformly distributed cases[31]. Humidity-driven reductions to the seasonal change in transmission in other locations will result in persistent epidemics, meaning cases will occur outside of the typical "RSV season" requiring changes to the temporal allocation of resources. Changing dynamics can also alter the age of infection[36,37]. This consideration is important given that early RSV infections have been implicated in the later life development of asthma[17].

Understanding the impact of climate change on the transmission dynamics of infectious diseases is critical for predicting and preparing for future outbreaks and the optimal deployment of future vaccines. Our results highlight the nonlinear impact of climate on RSV dynamics, across a wide range of climatic conditions. Shifts in the timing and magnitude of RSV outbreaks will be location-specific, depending on the realized climatic change. In regions where transmission is dominated by humidity, we can be relatively confident in our projections. However, in regions where precipitation dominates transmission, the future dynamics of RSV are more uncertain. Efforts to better resolve precipitation projections across climate models will improve the precision of these results. These findings also have potential implications for other airborne pathogens and future research efforts should be directed towards understanding the various processes through which climate affects airborne transmission. More broadly, our preliminary comparison between RSV and influenza reveals the potential for comparative studies to elucidate the mechanisms of climatic forcing for major respiratory infections.

## Methods
**Data**. The US county-level RSV dataset comes from hospitalizations data originally obtained from the State Inpatient Databases (SIDs) of the Healthcare Cost and Utilization Project (HCUP) maintained by the Agency for Healthcare Research and Quality (AHRQ). HCUP SIDs data represents on average 96% of community hospital inpatient discharges for reporting states, though not all states participate in data collection. There are some fluctuations in the number of community hospitals reporting over time, depending on the state. The standard deviation in percentage reporting in 2% on average. Hospitalization records that included the International Classification of Diseases 9th revision, Clinical Modification (ICD-9-CM) code for RSV (079.6, 466.11, 480.1) within the 15 discharge diagnoses were included. Hospitalization data were made available at the weekly level. Hospitalization data are available from 1989, however, a change in reporting codes in 1996 results in a large increase in cases at this time. To avoid potential bias from the switch in reporting, we take all county-level observations from 1997 onward. The longest

**a**

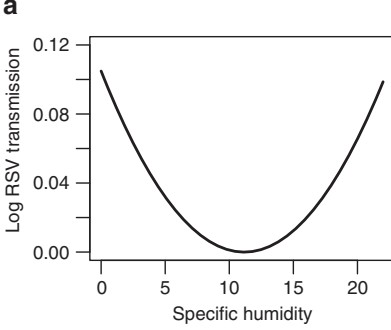

**b**

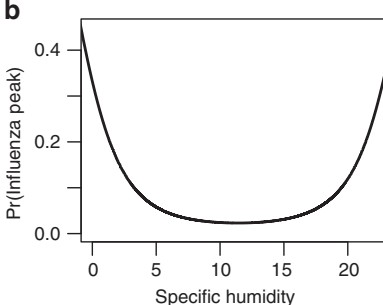

**Fig. 4** Comparison with influenza results. Removing precipitation from our regression model and including a quadratic humidity term reveals a very similar response (**a**) to earlier work on influenza (**b** adapted from Tamerius et al.[9], Fig 3a) suggesting potential similar mechanisms underlie the climate effect on the two diseases. Predicted minimum transmission for RSV occurs at 11.16 g/kg, with influenza found to be similarly 11–12 g/kg[9]. Tamerius et al. suggest precipitation may drive the right hand side of the humidity-influenza curve, as we find for RSV.

time series spans 1997–2011. We remove counties with very sparse observations (max incidence <10 cases) and we also remove series of zero observations from the start of the time series. We remove any counties that have <5 years of consecutive observations due to potential erroneous model-fitting over this shorter time series, though these counties are retained when plotting spatial averages.

State-level Mexico hospitalization data come from the Subsistema Automatizado de Egresos Hospitalarios (SAEH) collected by the Sistema Nacional de Información en Salud (SINAIS) and overseen by Secretaría de Salud. We take data from all cases of bronchiolitis (ICD-10 codes J21.0, J21.1, J21.8, J21.9), a serious respiratory tract infection primarily caused by RSV[38] (Supplementary Table 8). Data were summed over the week to match the US data and reflect the generation time of RSV. Mexico data cover all years from 2000 to 2014.

Temperature and specific humidity data come from the North America Regional Reanalysis (NARR) gridded dataset[39], produced by the National Centers for Environmental Prediction (NCEP). This high resolution dataset (32 km, daily observations) spans the geographic and temporal extent of our health data. Precipitation data come from Climate Hazards Group InfraRed Precipitation with Station data (CHIRPS)[40]. Spatial averages are constructed using shapefiles for US counties from the United States Census Bureau and for Mexican states.

Climate projections data come from the Coupled Model Intercomparison Project Phase 5 (CMIP5). Precipitation projections come from all CMIP5 models run under the Representative Concentration Pathway 8.5 scenario. Specific humidity is calculated from projections for temperature and relative humidity based on the multi-model mean, accessed via the KNMI Climate Explorer (https://climexp.knmi.nl/start.cgi).

Population data for the US were obtained from publicly-available combined files of United States Census Bureau data available via the National Bureau of Economic Research. US birth data were downloaded from the Centers for Disease Control. Demographic data for Mexico were obtained from the Instituto Nacional de Estadística y Geografía. Annual county (US) and state (Mexico) population estimates from these sources were interpolated at the weekly level using a cubic spline to avoid stepwise jumps at year transitions.

County-level data in the United States are noisier than state-level averages used in ref. [27], however, they provide some advantages. For instance, state-level time series for California show a dynamic transition from biennial outbreaks to annual outbreaks in 1999. At the county level we observe biennial cycles persisting from 1999 onward in California but with northern counties and southern counties out-of-phase (Supplementary Fig. 4). This suggests that using county-level data may provide a more accurate picture of location-specific dynamics.

**Definitions**. We define three types of dynamic pattern observed in the location-specific time series within the dataset. Shallow-trough dynamics are defined as when the annual mean minimum incidence for a particular location exceeds 5% of the mean maximum incidence. Biennial locations are defined using the significance of the biennial signal using the Lomb-Scargle periodogram. We use a significance value of 1% (all p-values under 5% are shown in the chloropleth Supplementary Fig. 3a, as well as the ratio of biennial to annual Fourier amplitudes in Supplementary Fig. 3b[27]). When running simulations for Fig. 3, we define chaotic regions as locations where the Lyapunov constant is positive.

To calculate mean onset week we first calculate mean incidence per week (i.e., each week averaged over all years for a given locations), and then normalize these values between 0 and 1. We define the start of the epidemic period as the 26th week of the year, approximately the first week of July. This period is chosen as it spans all epidemics within our data (Supplementary Fig. 6). We define onset as when normalized incidence exceeds 0.2, assuming this value is low enough to constitute onset but high enough to exceed random fluctuations in the data. Using normalized incidence, as opposed to incidence/population thresholds, allows to create a uniform measure across both US and Mexican datasets in spite of sampling differences between the two locations.

**Modeling approach**. Our modeling approach follows two steps[29]. We first use the time series TSIR model, a discrete time adaptation of the SIR model[14,41], to estimate the unobserved susceptible population for each location over time. Next, we use the estimated susceptible time series to construct an empirical transmission rate. The transmission rate is used as a dependent variable in a panel regression where the effect of climate is estimated.

**The TSIR model**. The TSIR model describes the number infected and susceptible individuals as a set of difference equations. The number of susceptible individuals is given by:

$$S_{t+1} = S_t + B_t - I_t + u_t \tag{1}$$

where $S_t$ and $I_t$ are the number of susceptible individuals and the number of infected individuals, respectively, and the time period, $t$, is the generation time for RSV approximated as 1 week. $B_t$ are births and $u_t$ is additive noise, with $E[u_t] = 0$. The susceptible population at each time step can be written as $S_t = \bar{S} + Z_t$, where $\bar{S}$ is the mean number of susceptible individuals in the population and $Z_t$ is the unknown deviation from the mean number of susceptible individuals at each time step. The susceptible equation can thus be rewritten in terms of deviations $Z_t$ and iterated successively with the starting condition $Z_0$, giving:

$$\sum_{k=0}^{t-1} B_k = -Z_0 + 1/\rho \sum_{k=0}^{t-1} Ir_k + Z_t + u_t \tag{2}$$

where $\rho$ is the reporting rate which accounts for both under-reporting of RSV hospitalizations as well as infections that did not result in hospitalization and $Ir_k$ is the reported incidence. Using this equation, $Z_t$ is estimated as the residuals from the linear regression of cumulative births on cumulative cases, assuming $u_t$ is small. The inverse of the slope of the regression line provides an estimate of the reporting rate $\rho$. $Z_t$ estimates can be used to reconstruct the susceptible time series though must be combined with an estimate for $\bar{S}$. We calculate this by defining the expected number of infected cases at each time step, $E[I_{t+1}]$, as:

$$E[I_{t+1}] = \frac{\beta_t I_t^\alpha S_t}{N_t} \tag{3}$$

which is log-linearized as:

$$\ln(E[I_{t+1}]) = \ln(\beta_t) + \alpha \ln(I_t) + \ln(\bar{S} + Z_t) - \ln(N_t) \tag{4}$$

where $\beta_t$ are biweekly factors that capture the seasonal trend in transmission rate and $\alpha$ is a constant that captures heterogeneities in mixing and the discretization of a continuous time process. We fix $\alpha$ at 0.97 to be consistent with prior studies[42]. Biweekly seasonal betas, as opposed to weekly betas, are estimated to avoid the overfitting of parameters due to the high correlation in transmission rates across successive weeks. Equation (4) is fit using a Poisson regression with log link. Final results are robust to using a negative binomial at this stage (Supplementary Table 7). The mean number of susceptible individuals, $\bar{S}$, can then be estimated using marginal profile likelihoods from estimating Eq. (4), for a range of candidate values. The TSIR model is used to reconstruct $S_t$ for each location in the dataset. An empirical estimate of the transmission rate, $Em\beta_t$, was then calculated for each location and time step in the dataset by rewriting Eq. (3), assuming that $E[I_{t+1}] = I_{t+1}$:

$$Em\beta_t = \frac{I_{t+1} N_t}{I_t^\alpha S_t} \tag{5}$$

To avoid inflation of $Em\beta_t$ we add one to zero observations in the infected time series which represents continual low-level background transmission resulting in the lack of epidemic extinction we observe in the data. Model results are also robust to removing zero observations from the infected time series (Supplementary Table 3). For fitting the TSIR we use the tsiR package[43]. Example TSIR fits are shown in Supplementary Fig. 7.

We remove data from all US counties where the $R^2$ of the TSIR fit is <0.5 (101 counties). These locations tend to be in counties with very low population numbers where low-level stochastic variability in cases becomes proportional to the size of seasonal variation. Our results are robust to using the full dataset and a population-based cut off (Supplementary Tables 5 and 6). The time series in the Mexican dataset are noisier than the United States, even for larger population states, which we hypothesize is due to the stochastic nature of rainfall drivers that dominate in this region and also due to sampling issues such as a higher threshold for hospitalization. In Mexico, we remove data from two states where the TSIR model does not provide a good fit due to very sparse data (under 10 cases at maximum in the time series): Colima and Queretaro. Our final dataset to which we fit the main regression model has 214 locations and a total of 119,802 location-by-week observations. Model results are robust to including data from all US counties and Mexico (Supplementary Table 5).

**Panel regression.** We fit a linear regression model using the empirical transmission rate as a dependent variable:

$$\ln(\mathrm{Em}\beta_{t,l}) = b_1(1/H_{t,l}) + b_2 P_{t,l} + \gamma_{l,m} + \delta_{l,y} + \epsilon_{t,l} \quad (6)$$

where $\mathrm{Em}\beta_{t,l}$ is the empirical transmission at time $t$ and location $l$, $H_{t_l}$ is humidity, and $P_{t,l}$ is precipitation. We include location-by-month dummies $\gamma_{l,m}$ that remove location-specific seasonal variation in transmission which may be confounded by other seasonally varying factors such as school semesters. We also include location-by-year dummies $\delta_{l,y}$ that remove location-specific trends in transmission or episodic transmission events that may be spuriously correlated with climate. Standard errors are clustered at the location level, though White's standard errors give similar results (Supplementary Table 4).

Nonparametric binned models and general additive models are also fitted to test the accuracy of the fitted functional form of the specific humidity and precipitation relationship (Supplementary Fig. 8 and 9). Lagged precipitation and humdity, in the week prior to transmission, are tested but not found to be significant (Supplementary Table 9).

**Simulations.** We run several different simulations to test for the effect of climate on the epidemic dynamics of RSV. To allow for climate to fully characterize the seasonal changes in transmission and mean transmission we re-estimate Eq. (6) removing dummy variable controls as:

$$\ln(\mathrm{Em}\beta_{t,l}) = b_1(1/H_{t,l}) + b_2 P_{t,l} + \epsilon_{t,l} \quad (7)$$

This model may be biased by other seasonally varying or location-specific trends and as such represents an upper bound on the climate effect. Results from this model are shown in Supplementary Fig. 11 and are significant ($p \ll 0.001$).

We run simulations over the parameter space of mean transmission and seasonal change in transmission values with the aim of evaluating (1) the location of bifurcations (Fig. 3a left plot) and (2) the difference between present and 2100 future projected dynamics (Fig. 3a right plot). In order to smoothly vary the seasonal change in transmission across the parameter space we use a cosine function to represent seasonality in humidity. Simulations are run for 50 years to remove the effect of transients and only the last 10 years are analyzed. To keep other factors constant, demographics in these simulations are based on Kings County, New York. In Fig. 3a (right), we plot an arrow for each location within our dataset where the base of the arrow represents the estimated present parameter values and the point of the arrow represents the projected future parameter values.

To produce Fig. 3b we run simulations this time allowing seasonal climate variations to fully specify the seasonal and mean transmission rate, removing the cosine structure. We take output from all 23 climate models included in the CMIP5 mean. For each model we calculate the ratio of the week-of-year average rainfall at the end of the century to the start of the century (using a 5-year average anomaly in each case). We then apply this ratio to the observed precipitation in our dataset also averaged over a 5-year period to minimize bias from comparing model output to observational data. We apply the same method to the model-mean specific humidity. Projected precipitation and humidity are input into Eq. (7) to calculate a seasonally varying transmission rate. Forward simulations are run for 50 years to remove transients with only the last 5 years shown in Fig. 3b.

**Reporting summary.** Further information on research design is available in the Nature Research Reporting Summary linked to this article.

## Data availability

The US RSV hospitalization data are available from the Agency for Healthcare Research and Quality upon signing a data use agreement. The Mexico hospitalization dataset is available from the Sistema Nacional de Información en Salud.

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

## Acknowledgements

V.E.P. is supported by grant R01AI137093 from the US National Institutes of Health/ National Institute of Allergy and Infectious Diseases. This study is supported by the Cooperative Institute for Climate Science, the Princeton Institute for International and Regional Studies and the Princeton Environmental Institute.

## Author contributions

Conceptualization: R.E.B., A.S.M., G.V., C.J.E.M., and B.T.G.; data curation: C.V. and W.Y.; formal analysis, software, and visualization: R.E.B.; methodology: R.E.B., A.S.M., C.E.W., V.E.P., C.V., G.V., C.J.E.M., and B.T.G.; writing, original draft: R.E.B.; writing, reviewing and editing: R.E.B., A.S.M., C.E.W., V.E.P., C.V., G.V., C.J.E.M., and B.T.G.

## Competing interests

The authors declare no competing interests.
