## [Peer Review File · Nature Communications]

Reviewers' Comments:

Reviewer #1:

Remarks to the Author:

This paper has a good premise, is generally well written, and presents some interesting findings vis-à-vis the effects of climate on RSV. It merits publication in Nature Communications following revision. The main problem I had with the manuscript was in the presentation of the Methods, which are incomplete and confusing. The authors need to re-work this section, and expand them in the Supplement if necessary.

Line 39: I would not say 'defining our ability to predict...' that is too strong—it often only defines conditions favorable for transmission and thus seasonality, not the specific timing and magnitude. Maybe change to 'often delineating our ability to predict...'

Lines 49-51: This is a bit awkward. Maybe change to 'exhibit more regular dynamics making the long term...'

Line 75: Change 'forecast' to 'project'

Lines 84-86: I don't follow the logic here. Could there not be some 'seasonal' behavioral factor—time indoors in air conditioning—occurring at both places? Please explain and amend the text to clarify.

Line 132: 'swing' seems too loose and imprecise. How about 'has large changes in predicted transmission'? Same for Line 147 (larger seasonal changes) and the caption to Figure 2 and elsewhere. (Bottom line, I don't like 'swing'). Maybe 'shift'...

Line 164: The words after the semi-colon are not an independent clause (no verb), so change the semi-colon to a colon or ', and'.

Line 246-247, 254: 'data were', 'data are', 'data come'

Line 271: 'data in the USA are', '; however, it provides'

Line 284: 'is shown in'

Lines 287-289: This isn't clear. To calculate mean onset, calculate mean incidence per week (i.e. each week averaged over all years for a given location), then normalize these values to between 0 and 1. I find 'averaging over the whole time series within a particular location' confusing. Plus, with 'this value' it sounds like the authors averaged to a single value and then tried to normalize, which is meaningless.

Line 291: 'this value is'

Lines 294-305: How is N_t obtained, i.e. what is the data source? And how does it vary through time? Also, how complete are the state-level HCUP data? Is it all hospitalizations or only a partial sample within each county (where there may be more than 1 hospital)? And do the hospitals change in capacity through time, which could be a problem? Further, and even though these issues are partially addressed in lines 306-326, it would also be good to discuss the scaling when talking about the HCUP data. Specifically, how are the count data from HCUP reconciled with the population data—is this solely through the estimate of ρ ? If it is supposed to represent hospitalized RSV rates does ρ scaling account for all infections (both hospitalized and otherwise)?

Equation 3: I don't see how this is a re-writing of the susceptible equation—what is the substitution here? And define r_k . And what is the data source for births (so that the regression can be performed and ρ estimated)? And the residuals are technically $Z_t + u_t$ not Z_t .

Equation 5: The t subscript for Beta_t has been dropped for Beta in moving from Equation 4. As Beta_t is the quantity being estimated for the main analysis, there is something circular and squirrely in this substitution. Also why is it 'biweekly' (when the data are weekly, and there should be only one estimate of $\ln(\text{beta})$, which is the estimated intercept of the regression, no)? And how is it a 'seasonal trend' if it is one value? I like this manuscript, but you can tell I am finding a lot wanting in the Methods description.

Line 354: 'run simulations'

Lines 354-355: The terms 'mean log transmission' and 'amplitude of seasonal change in log transmission' are not clear. The rationale for this first set of simulation is not clear either. And the projected (not predicted) amplitudes don't mean much unless compared to estimates for the present. Note, the final paragraph is very clear, but this next to last one is not communicative.

Figure S1—are these histograms of daily climatology? Monthly? Please include this in the figure caption.

Figure S5 – needs color scale bars indicating latitude-color relationship.

Tables S2-S6 – are the numbers in parentheses standard error? Also, Table S6, include the lag value (month?) in the caption.

Reviewer #2:

Remarks to the Author:

Review of Dissecting climatic influences on nonlinear epidemic dynamics: the current and future epidemiology of respiratory syncytial virus

This is an interesting paper, on a well-motivated topic; as the authors point out, the effect of climate on the transmission of diseases other than influenza (and DENV) has been on the whole poorly studied.

I have several concerns with this analysis, however:

1. There are over 3,000 counties in the US but in most of the results the authors focus on only four for some reason, and it is unclear why they chose those four in particular.
2. The authors state in passing that timing of school holidays cannot be responsible for the patterns observed. But different states have different start dates for the school year (some starting in August and some starting in September, and different schools have different timing of the spring break). Cauchemez et al (2008) found that the timing of school holidays significantly effects patterns of influenza incidence, thus it would be very surprising if the same were not true of RSV (see <https://www.nature.com/articles/nature06732>). The authors should incorporate in their analysis at least a state-level factor that accounts for the transmission difference due to the start dates of the school year in the temporal analysis.
3. While the authors use Fourier methods to attempt to detect annual and biennial patterns, such methods do not yield a p-value for the significance of the effect, and thus it may superficially appear

that there is an annual or biennial cycle, but the p-value is actually not significant. Overall, the methodology the authors use to assess the presence of annual and biennial patterns is ad hoc. Instead, it would be better to use methods such as Lomb-Scargle periodograms, or, even more preferable for use with count data, Negative Binomial harmonic linear regression, regressing the log of the weekly incidence on sine and cosine terms with period of one year, and period of two years. For more information on harmonic (trigonometric) regression, see <https://academic.oup.com/biomet/article-abstract/77/1/1/271047?redirectedFrom=fulltext> and/or <https://journals.plos.org/plosone/article?id=10.1371/journal.pone.0205151>

4. Once p-values assessing the significance of annual and biennial patterns has been incorporated into the analysis, a choropleth plot of counties showing the significance of biennial and annual effects would be helpful in visualizing the geospatial patterns.
5. The authors use a Susceptible-Infected-Recovered model, but the incubation period for RSV is not negligible, being around four to five days (see <https://www.sciencedirect.com/science/article/abs/pii/S1473309909700696>). In addition, it is unclear why the authors did not cite the Moore et al (2012) paper, which uses a more appropriate SEIRS model with seasonal transmission to model RSV outbreaks, since it is directly relevant to this work: <https://journals.plos.org/plosone/article?id=10.1371/journal.pone.0100422>
6. The authors give few details about the fitting methods used. Negative Binomial likelihood methods are most appropriate for count data, but it is unclear what methods were used.
7. Line 109 the authors state: "α captures heterogeneities in mixing and the effects of discretization.", and then later on line 303 they state that they fix it to 0.97 "to be consistent with prior studies". The prior studies are not cited, and it is unclear to me how alpha corrects for heterogeneities in mixing, and/or corrects for discretization.
8. Line 318: the authors state that beta captures the seasonal trend in transmission rate and includes both social and environmental drivers. They also state that biweekly factors are used to avoid over-fitting of seasonal effects. It is unclear what is meant by either of these statements.
9. The authors state on line 327: "We remove data from all US counties where the R² of the TSIR fit is less than 0.5 (101 counties). These locations tend to be in counties with very low population numbers where low-level stochastic variability in cases becomes proportional to the size of seasonal variation." This selection biases the analysis to only look at locations that fit the authors' hypothesis of annual or biennial variation. Much more reasonable would be to simply exclude all counties with population under some nominal cutoff in population (such as 100,000).

Reviewers' comments:

Reviewer #1 (Remarks to the Author):

This paper has a good premise, is generally well written, and presents some interesting findings vis-à-vis the effects of climate on RSV. It merits publication in Nature Communications following revision. The main problem I had with the manuscript was in the presentation of the Methods, which are incomplete and confusing. The authors need to re-work this section, and expand them in the Supplement if necessary.

We thank the reviewer for their helpful comments on the paper. We have made several changes to the paper, including updating the presentation of the methods section, details are below:

Line 39: I would not say 'defining our ability to predict...' that is too strong—it often only defines conditions favorable for transmission and thus seasonality, not the specific timing and magnitude. Maybe change to 'often delineating our ability to predict...'

Thank you, this has now been updated following the reviewer's suggestion. Line 39.

Lines 49-51: This is a bit awkward. Maybe change to 'exhibit more regular dynamics making the long term...'

Updated as per the reviewer's suggestion. Line 51.

Line 75: Change 'forecast' to 'project'

Updated as per the reviewer's suggestion. Line 76.

Lines 84-86: I don't follow the logic here. Could there not be some 'seasonal' behavioral factor—time indoors in air conditioning—occurring at both places? Please explain and amend the text to clarify.

We thank the reviewer for this comment. This section of the text has now been updated (lines 86-90) based on comments from both reviewers. We aim to make the point that the latitudinal gradient in onset timing is unlikely to be driven by non-climatic factors alone, in particular the timing of school semesters. Time spent indoors due to air-conditioning could also be considered an indirect climatic effect.

New text "The earliest onset of the epidemic occurs in the summer months simultaneously in Miami, Florida and the Yucatan peninsula in Mexico. Onset timing then follows a latitudinal gradient northward such that onset occurs last in the northern US in late December (Fig. 1B, Fig. S5). The timing of school semesters is unlikely to drive this gradient: while US school start dates vary by state, in Mexico the first day of the school year is standardized across the country."

Line 132: 'swing' seems too loose and imprecise. How about 'has large changes in predicted transmission'? Same for Line 147 (larger seasonal changes) and the caption to Figure 2 and elsewhere. (Bottom line, I don't like 'swing'). Maybe 'shift'...

We thank the reviewer for this suggestion. Uses of "swing" have now been updated to "changes" and "transmission swing" is now "seasonal transmission change". We also updated Fig. 3A and 3C plot axes to "seasonal transmission change". We previously used "change" to describe the 2100-2010 *difference* in seasonal transmission change. This has now been updated to "2100-2010 difference" in Fig. 3C and lines 191-197 to minimize confusion.

Line 164: The words after the semi-colon are not an independent clause (no verb), so change the semi-colon to a colon or ', and'.

Corrected as per the reviewer's suggestion. Line 172.

Line 246-247, 254: 'data were', 'data are', 'data come'

Corrected as per the reviewer's suggestion as well as in other places in the text where a similar error exists. Line 260.

Line 271: 'data in the USA are', '; however, it provides'

Corrected as per the reviewer's suggestion. Line 293.

Line 284: 'is shown in'

Corrected as per the reviewer's suggestion – though sentence now removed in this updated paragraph.

Lines 287-289: This isn't clear. To calculate mean onset, calculate mean incidence per week (i.e. each week averaged over all years for a given location), then normalize these values to between 0 and 1. I find 'averaging over the whole time series within a particular location' confusing. Plus, with 'this value' it sounds like the authors averaged to a single value and then tried to normalize, which is meaningless.

We thank the reviewer for pointing out this unclear description. It is now updated. Lines 310-312.

New text "To calculate mean onset week we first calculate mean incidence per week (i.e. each week averaged over all years for a given locations), and then normalize these values between 0 and 1."

Line 291: 'this value is'

Corrected as per the reviewer's suggestion. Line 315.

Lines 294-305: How is N_t obtained, i.e. what is the data source? And how does it vary through time? Also, how complete are the state-level HCUP data? Is it all hospitalizations or only a partial sample within each county (where there may be more than 1 hospital)? And do the hospitals change in capacity through time, which could be a problem? Further, and even though these issues are partially addressed in lines 306-326, it would also be good to discuss the scaling when talking about the HCUP data. Specifically, how are the count data from HCUP reconciled with the population data—is this solely through the estimate of ρ ? If it is supposed to represent hospitalized RSV rates does ρ scaling account for all infections (both hospitalized and otherwise)?

Population data was obtained from publicly-available combined files of United States Census Bureau data available via the National Bureau of Economic Research <https://data.nber.org/data/census-intercensal-county-population.html>. Annual data is interpolated using a cubic spline to avoid stepwise jumps between years. This information is now added to the data description, line 286-291.

HCUP SIDS data represents on average 96% of community hospital inpatient discharges for reporting states, though not all states report in all time periods. Based on available data, the lowest state-level coverage for reporting states is 94.7% and the highest is 97.1%. There are some small fluctuations in the number of community hospitals reporting data over time, depending on the state. The standard deviation in percentage dataset coverage (in terms reporting community hospitals compared to total community hospitals) ranges from 0 (meaning the same number of hospitals report at every time point) to 10% in Rhode Island, with a mean of 2%. This information is now added to the HCUP data description in lines 253-257.

ρ , the reporting rate, accounts for both under-reporting of hospitalizations for RSV, as well as infections that did not result in hospitalization. This is now clarified in lines 345-346.

New text:

"HCUP SIDS data represents on average 96% of community hospital inpatient discharges for reporting states, though not all states participate in data collection. There are some fluctuations in the number of community hospitals reporting over time, depending on the state. The standard deviation in percentage reporting is 2% on average."

"Population data for the US were obtained from publicly-available combined files of United States Census Bureau data available via the National Bureau of Economic Research \cite{NBERpop}. US birth data were downloaded from the Centers for Disease Control \cite{CDC}. Demographic data for Mexico were obtained from the Instituto Nacional de Estadística y Geografía. Annual county (US) and state (Mexico) population estimates from these sources were interpolated at the weekly level using a cubic spline to avoid stepwise jumps at year transitions.

"where ρ is the reporting rate which accounts for both under-reporting of RSV hospitalizations as well as infections that did not result in hospitalization."

Equation 3: I don't see how this is a re-writing of the susceptible equation—what is the substitution here? And define ρ . And what is the data source for births (so that the regression can be performed and ρ estimated)? And the

residuals are technically $Z_t + u_t$ not Z_t .

We thank the reviewer for pointing out this issue with the presentation of the methodology. The susceptible equation is first rewritten in terms of deviations Z_{t-1} and then iterated from the starting condition Z_{t-0} to get equation (3). A new sentence has been added to clarify this step "The susceptible equation can thus be rewritten in terms of deviations Z_{t-1} and iterated successively with the starting condition Z_{t-0} ." Lines 343-344.

$I_{r,t}$ is the reported incidence. Now defined in text line 346.

u_{t-1} is assumed to be small, making the residuals Z_{t-1} . Lines 348.

Equation 5: The t subscript for β_t has been dropped for β in moving from Equation 4. As β_t is the quantity being estimated for the main analysis, there is something circular and squirrely in this substitution. Also why is it 'biweekly' (when the data are weekly, and there should be only one estimate of $\ln(\beta)$, which is the estimated intercept of the regression, no)? And how is it a 'seasonal trend' if it is one value? I like this manuscript, but you can tell I am finding a lot wanting in the Methods description.

We apologize for the error in the presentation of the methods: β should read β_t . This is now corrected. Lines 353, equation 4.

We have now updated the methods section to attempt to clarify the steps taken. We summarize the procedure in new text (Lines 319-325): ".our modelling approach follows two steps. We first use the time series Susceptible-Infected-Recovered model (TSIR), a discrete time adaptation of the SIR model to estimate the unobserved susceptible population for each location over time. Next, we use the estimated susceptible time series to construct an empirical transmission rate. The transmission rate is used as a dependent variable in a panel regression where the effect of climate is estimated." The ordering of the methods has now been changed to reflect these steps.

Biweekly seasonal factors, as opposed to weekly factors are estimated in the GLM regression stage to avoid the overfitting of parameters due to the high correlation in transmission rates across successive weeks. This sentence is now added in lines 357-359.

We have now clarified in the text the distinction between empirical β , now defined as β_{em} and the seasonal β s fitted in the TSIR GLM regression, β_t (Lines 364, 341, Equation (5)). We use seasonal β s, β_t , only for the purpose of estimating the mean number of susceptibles, S_{bar} . Given our identification of the climate effect comes from deviations from mean transmission, the value of S_{bar} will have minimal effect on our climate results. While the empirical β is an approximation of the true transmission rate, earlier simulation work has shown that this approach is able to retrieve an unbiased estimates of climate dependence (Baker, Mahmud and Metcalf 2018).

Line 354: 'run simulations'

Corrected as per the reviewer's suggestion. Line 399.

Lines 354-355: The terms 'mean log transmission' and 'amplitude of seasonal change in log transmission' are not clear. The rationale for this first set of simulation is not clear either. And the projected (not predicted) amplitudes don't mean much unless compared to estimates for the present. Note, the final paragraph is very clear, but this next to last one is not communicative.

We thank the reviewer for pointing out the confusing wording in this paragraph. The paragraph has now been rewritten:

"We run simulations over the parameter space of mean transmission and seasonal change in transmission values with the aim of evaluating 1) the location of bifurcations (Fig. 3A left plot) and 2) the difference between present and 2100 future projected dynamics (Fig. 3A right plot). In order to smoothly vary the seasonal change in transmission across the parameter space we use a cosine function to represent seasonality in humidity. Simulations are run for 50 years to remove the effect of transients and only the last 10 years are analysed. To keep other factors constant, demographics in these simulations are based on Kings County, New York. In Fig. 3A (right) we plot an arrow for each location within our dataset where the base of the arrow represents the estimated present parameter values and the point of the arrow represents the projected future parameter values." Lines 399-417.

Fig. 3. has also been updated to be consistent with the parameter definitions used earlier in the text.

Figure S1—are these histograms of daily climatology? Monthly? Please include this in the figure caption.

Weekly. This has now been added to the caption.

Figure S5 – needs color scale bars indicating latitude-color relationship.

This has now been updated, as per the reviewer's suggestion. Note, this is now Fig. S6.

Tables S2-S6 – are the numbers in parentheses standard error? Also, Table S6, include the lag value (month?) in the caption.

We thank the reviewer for highlighting unclear labelling. The parentheses show the standard errors. We have now added this to the caption for all tables. In S6 (now S7), climate variables are lagged by one week. This is now added to the caption.

Reviewer #2 (Remarks to the Author):

Review of Dissecting climatic influences on nonlinear epidemic dynamics: the current and future epidemiology of respiratory syncytial virus

This is an interesting paper, on a well-motivated topic; as the authors point out, the effect of climate on the transmission of diseases other than influenza (and DENV) has been on the whole poorly studied.

We thank the reviewer for their helpful comments on the paper. Detailed responses are below:

I have several concerns with this analysis, however:

1. There are over 3,000 counties in the US but in most of the results the authors focus on only four for some reason, and it is unclear why they chose those four in particular.

We thank the reviewer for pointing out the lack of clarity in the results presentation. Because of the size of our dataset, our aim was to provide both synoptic results across locations (i.e. Fig. 2A, Fig. 2E, Fig. 3A, Fig. 3C) and specific results from locations with exemplar dynamics (Fig. 2B-D, Fig. 3B). This is now clarified in lines 131-139. We also provide results from additional locations in the Supporting Information.

New text “We consider the implications of our estimated climate effect for four locations in our dataset with exemplar dynamic patterns and timing of epidemic onset (Fig. 2B, Fig. 2C). In Fig. 2D we show the seasonal trajectories for these locations, with the mean monthly rainfall and humidity overlaid on our predicted transmission effect. Hennepin County, MN, an example of a location with biennial outbreaks, has a large change in predicted transmission over the course of the year, with maximum transmission in the winter months (Fig. 2C) when peak incidence also occurs (Fig. 2B). Kings County, NY, an example of a location with annual outbreaks, has a wintertime peak in predicted transmission driven by declining humidity, though not to the same extent as the biennial test case. In contrast, our model predicts that Miami-Dade County, FL, an example of a “shallow-trough” location, experiences peak transmission in the summer months driven by summertime rainfall.”

2. The authors state in passing that timing of school holidays cannot be responsible for the patterns observed. But different states have different start dates for the school year (some starting in August and some starting in September, and different schools have different timing of the spring break). Cauchemez et al (2008) found that the timing of school holidays significantly affects patterns of influenza incidence, thus it would be very surprising if the same were not true of RSV (see <https://www.nature.com/articles/nature06732>). The authors should incorporate in their analysis at least a state-level factor that accounts for the transmission difference due to the start dates of the school year in the temporal analysis.

We thank the reviewer for highlighting this issue. The mean age of infection of RSV is very young, under 2 years, so it is less clear that schooling is a driver. Furthermore, while US counties have different school timings, in Mexico school semesters are standardized. Despite this, we see a strong latitudinal gradient in RSV epidemic onset in Mexico (Fig. S5). This makes it unlikely that school openings alone can explain the onset timing. The text is now updated in lines 84-90.

New text “The earliest onset of the epidemic occurs in the summer months simultaneously in Miami, Florida and the Yucatan peninsula in Mexico. Onset timing then follows a latitudinal gradient northward such that onset occurs last in the northern US in late December (Fig. 1B, Fig. S5). The timing of school semesters is unlikely to drive this gradient: while US school start dates vary by state, in Mexico the first day of the school year is standardized across the country.”

3. While the authors use Fourier methods to attempt to detect annual and biennial patterns, such methods do not yield a p-value for the significance of the effect, and thus it may superficially appear that there is an annual or biennial cycle, but the p-value is actually not significant. Overall, the methodology the authors use to assess the presence of annual and biennial patterns is ad hoc. Instead, it would be better to use methods such as Lomb-Scargle periodograms, or, even more preferable for use with count data, Negative Binomial harmonic linear regression, regressing the log of the weekly incidence on sine and cosine terms with period of one year, and period of two years. For more information on harmonic (trigonometric) regression, see <https://academic.oup.com/biomet/article-abstract/77/1/1/271047?redirectedFrom=fulltext> and/or <https://journals.plos.org/plosone/article?id=10.1371/journal.pone.0205151>

We thank the reviewer for this helpful suggestion. Originally, we followed the methods in (Pitzer et al 2015) when detecting bienniality using Fourier methods, however, we have now updated our plots to use the significance of the Lomb-Scargle periodogram as the test for bienniality. Lines 300-307. Our main conclusions are unaffected by the update, though more rigorously supported by the change in method.

New text “Biennial locations are defined using the significance of the biennial signal using the Lomb-Scargle periodogram. We use a significance value of 1% (all p-values under 5% are shown in the choropleth Fig. S3A, as well as the ratio of biennial to annual fourier amplitudes in Fig. S3B (Pitzer 2015))”

4. Once p-values assessing the significance of annual and biennial patterns has been incorporated into the analysis, a choropleth plot of counties showing the significance of biennial and annual effects would be helpful in visualizing the geospatial patterns.

We now include the choropleth plot in the Supplement, Fig. S3A.

5. The authors use a Susceptible-Infected-Recovered model, but the incubation period for RSV is not negligible, being around four to five days (see <https://www.sciencedirect.com/science/article/abs/pii/S1473309909700696>). In addition, it is unclear why the authors did not cite the Moore et al (2012) paper, which uses a more appropriate SEIRS model with seasonal transmission to model RSV outbreaks, since it is directly relevant to this work: <https://journals.plos.org/plosone/article?id=10.1371/journal.pone.0100422>

We thank the reviewer for drawing our attention to the suggested paper, which was indeed relevant and an omission from our earlier literature review. We have now cited the paper lines 58.

Our aim was to develop the simplest model that could capture both observed dynamics (e.g. biennial cycles) while allowing for the estimation of a weekly, season-specific transmission rate to compare with climate drivers. Therefore, we did not consider age-structure, RSV sub-strains or secondary infections. This is now clarified in the discussion, line 223. While there may be some return to susceptibility following an RSV infection, the observed signature of multiannual dynamics and therefore herd immunity suggests this is not a return to full susceptibility, making the TSIR a reasonable approximation (Pitzer et al 2015, Morris et al 2018).

The TSIR model has been shown to be robust in terms of capturing overall disease dynamics and the bifurcation diagram for diseases with incubation periods (Bjornstad et al 2002, Metcalf et al 2009).

6. The authors give few details about the fitting methods used. Negative Binomial likelihood methods are most appropriate for count data, but it is unclear what methods were used.

We thank the reviewer for pointing out the lack of clarity in the presentation of the methods. Negative binomial has been proposed for fitting the TSIR model (see discussion in Bjornstad et al. 2002), however, the choice of error distribution has not been shown to matter much for results (Bjornstad et al 2002, Takahashi et al 2016). For consistency with previous studies we use a Poisson GLM model with a log link. This is now clarified in lines 359.

New text “Equation 4 is fit using a Poisson regression with log link.”

7. Line 109 the authors state: “ α captures heterogeneities in mixing and the effects of discretization.”, and then later on line 303 they state that they fix it to 0.97 “to be consistent with prior studies”. The prior studies are not cited, and it is unclear to me how α corrects for heterogeneities in mixing, and/or corrects for discretization.

We thank the reviewer. The correct citation for α has now been included. Line 357.

New text “ α is a constant that captures heterogeneities in mixing and the discretization of a continuous time process. We fix α at 0.97 to be consistent with prior studies (Glass et al 2003).”

8. Line 318: the authors state that beta captures the seasonal trend in transmission rate and includes both social and environmental drivers. They also state that biweekly factors are used to avoid over-fitting of seasonal effects. It is unclear what is meant by either of these statements.

We thank the reviewer for pointing out this unclear section of the methods. Based on queries from both reviewers, the methods section has now been significantly updated to improve clarity.

There was an error in the original presentation of the methods, such that beta in equation (4) should have been β_t . This has been updated.

We have now clarified in the text the distinction between empirical beta, now defined as β_t and the seasonal betas fitted in the TSIR GLM regression, β_t . Line 364, 366, equation 5.

Biweekly seasonal factors, as opposed to weekly factors, are estimated to avoid the overfitting of parameters due to the high correlation in transmission rates across successive weeks. This sentence has now been added to lines 357.

9. The authors state on line 327: “We remove data from all US counties where the R^2 of the TSIR fit is less than 0.5 (101 counties). These locations tend to be in counties with very low population numbers where low-level stochastic variability in cases becomes proportional to the size of seasonal variation.” This selection biases the analysis to only look at locations that fit the authors’ hypothesis of annual or biennial variation. Much more reasonable would be to simply exclude all counties with population under some nominal cutoff in population (such as 100,000).

We thank the reviewer for this query. We have now included Table S5 where we remove data from counties with a population less than 500,000. The conclusions are similar to our main results.

References

Baker, Rachel E., Ayesha S. Mahmud, and C. Jessica E. Metcalf. "Dynamic response of airborne infections to climate change: predictions for varicella." *Climatic change* 148.4 (2018): 547-560.

Becker, Alexander D., and Bryan T. Grenfell. "tsiR: An R package for time-series Susceptible-Infected-Recovered models of epidemics." *PLoS one* 12.9 (2017): e0185528.

Bjørnstad, Ottar N., Bärbel F. Finkenstädt, and Bryan T. Grenfell. "Dynamics of measles epidemics: estimating scaling of transmission rates using a time series SIR model." *Ecological monographs* 72.2 (2002): 169-184.

Glass, Kathryn, Yincun Xia, and Bryan T. Grenfell. "Interpreting time-series analyses for continuous-time biological models—measles as a case study." *Journal of theoretical biology* 223.1 (2003): 19-25.

Metcalf, C. Jessica E., et al. "Seasonality and comparative dynamics of six childhood infections in pre-vaccination Copenhagen." *Proceedings of the Royal Society B: Biological Sciences* 276.1676 (2009): 4111-4118.

Morris, Sinead E., et al. "Demographic buffering: titrating the effects of birth rate and imperfect immunity on epidemic dynamics." *Journal of the Royal Society Interface* 12.104 (2015): 20141245.

Pitzer, Virginia E., et al. "Environmental drivers of the spatiotemporal dynamics of respiratory syncytial virus in the United States." *PLoS pathogens* 11.1 (2015): e1004591.

Takahashi, Saki, et al. "Hand, foot, and mouth disease in China: modeling epidemic dynamics of enterovirus serotypes and implications for vaccination." *PLoS medicine* 13.2 (2016): e1001958.

Reviewers' Comments:

Reviewer #1:

Remarks to the Author:

The authors have addressed my concerns and clarified the methodological presentation. I have a few minor comments below, but feel the paper should be accepted.

Line 124: Fig S9 should be Fig. S10, I believe.

Line 139: change 'a large' to 'large'

Line 293: 'it' refers to 'data', which is plural. Change 'it provides' to 'they provide'

Figure 3 and in the main text, the authors report the change in the amplitude of the seasonal cycle as 'seasonal transmission change'. It would be good to define 'log seasonal transmission change' somewhere (perhaps in the figure caption)./

Reviewer #2:

Remarks to the Author:

Review of Dissecting climatic influences on nonlinear epidemic dynamics: the current and future epidemiology of respiratory syncytial virus

I am happy to see the authors are now using Lomb-Scargle rather than Fourier methods for assessment of periodicity. However, I have some concerns that have not been addressed (all of which are associated with over-estimation of the statistical power of the analysis):

1. In my previous comments, I brought up the fact that the Poisson regression methods the authors use are not appropriate for over-dispersed data. And virtually all data are over-dispersed. The authors cite Bjornstad et al (2002) as a paper that they purport states that the error function used in the NB model "has not been shown to matter much". To the contrary, the paper the authors cite properly used Negative Binomial regression methods, and nowhere in the paper did those authors state that Poisson methods were preferable. The authors of this paper state they used Poisson regression methods to be consistent with previous work, but in fact those methods are inconsistent with previous work.

The way the authors can truly show it doesn't much matter is to repeat their analysis using Negative Binomial methods (using error function of their choosing) and compare the results to that of the Poisson regression model; I would be extraordinarily surprised if the uncertainty on the parameter estimates is the same (because I have yet to do an analysis of incidence data where the data were not significantly over-dispersed). The Negative Binomial confidence intervals will almost certainly be much larger, and the parameter in the Negative Binomial regression that takes into account over-dispersion in the data will likely not be consistent with zero.

2. This brings me to an important point which I did not bring up on prior review, but I have just noticed (and I apologise I did not catch this on the first version). The authors are using regression methods to estimate Emb_{β_t} , yet it appears they just use the central value of that quantity in the climate regression in Equation 6. I do not see a discussion in their methodology how uncertainties in the estimate of Emb_{β_t} are incorporated into the analysis (or even if this is considered at all). Lewis and Linzer (2005) discuss the heteroskedasticity problems that arise with regressing a dependent variable that itself is an estimate:

<https://pdfs.semanticscholar.org/0575/172b931858d6a713f8f10d41ae8ad9104edb.pdf>

The problem is that not taking into account the uncertainty on Emb_{t} can lead to over-estimation of the power of the analysis in detecting climate-dependent effects. An iterative weighted linear regression method such as the one Lewis and Linzer suggest in Section 4.2 of their paper should be used to get unbiased estimates of the confidence intervals on the regression parameter estimates. It is worth noting here that if the uncertainty on Emb_{t} is σ , the uncertainty on $\log(\text{Emb}_{\text{t}})$ is $w = \sigma / \text{Emb}_{\text{t}}$, and $1/w$ should be the weight used in the log link linear regression.

3. In my original review, I expressed concern that the authors were removing data where the R^2 was less than 0.5. As I stated in the original review, this likely biases the analysis results by over-inflating the estimated power. The authors state in their replies they have switched to a population cut-off instead, but the text of the revised version indicates they are still making this R^2 selection.

Reviewers' comments:

Reviewer #1 (Remarks to the Author):

The authors have addressed my concerns and clarified the methodological presentation. I have a few minor comments below, but feel the paper should be accepted.

We thank the reviewer for their positive comments about the manuscript.

Line 124: Fig S9 should be Fig. S10, I believe.

This has been corrected. Line 121.

Line 139: change 'a large' to 'large'

This has been corrected. Line 135

Line 293: 'it' refers to 'data', which is plural. Change 'it provides' to 'they provide'

This has been corrected. Line 286

Figure 3 and in the main text, the authors report the change in the amplitude of the seasonal cycle as 'seasonal transmission change'. It would be good to define 'log seasonal transmission change' somewhere (perhaps in the figure caption)./

Thank you, this has now been added to the figure caption (Fig. 3).

New text "log seasonal transmission change (annual max - min transmission values)" (Fig. 3 caption).

Reviewer #2 (Remarks to the Author):

Review of Dissecting climatic influences on nonlinear epidemic dynamics: the current and future epidemiology of respiratory syncytial virus

I am happy to see the authors are now using Lomb-Scargle rather than Fourier methods for assessment of periodicity. However, I have some concerns that have not been addressed (all of which are associated with over-estimation of the statistical power of the analysis):

1. In my previous comments, I brought up the fact that the Poisson regression methods the authors use are not appropriate for over-dispersed data. And virtually all data are over-dispersed. The authors cite Bjornstad et al (2002) as a paper that they purport states that the error function used in the NB model "has not been shown to matter much". To the contrary, the paper the authors cite properly used Negative Binomial regression methods, and nowhere in the paper did those authors state that Poisson methods were preferable. The authors of this paper state they used Poisson regression methods to be consistent with previous work, but in fact those methods are inconsistent with previous work.

The way the authors can truly show it doesn't much matter is to repeat their analysis using Negative Binomial methods (using error function of their choosing) and compare the results to that of the Poisson regression model; I would be extraordinarily surprised if the uncertainty on the parameter estimates is the same (because I have yet to do an analysis of incidence data where the data were not significantly over-dispersed). The Negative Binomial confidence intervals will almost certainly be much larger, and the parameter in the Negative Binomial regression that takes into account over-dispersion in the data will likely not be consistent with zero.

We thank the reviewer for following up on this point. As they recommend, we now present results obtained from analyzing the data using the Negative Binomial model for the GLM component in the Supplementary Materials. These results are included for the whole sample in Table S7 (and reproduced below). We note in the text that using a negative binomial model does not significantly change the results relative to our climate estimates. Indeed, in our approach, the GLM regression is only used to estimate S_{bar} , the mean number of susceptibles. Since our climate effects are estimated based on deviations from a local mean empirical beta around S_{bar} , the major impact of using a negative binomial framing, i.e., changing S_{bar} is unlikely to modify the results relative to climate effects. We choose to use a Poisson regression for consistency with the other TSIR papers.

	Dependent variable:		
	Empirical Beta		
	(USA + MEX)	(USA)	(MEX)
1/Specific Humidity	1.886e-04*** (2.381e-05)	1.861e-04*** (2.424e-05)	1.391e-04 (1.150e-04)
Precipitation	1.000e-03** (4.642e-04)	3.174e-04 (4.481e-04)	7.427e-03*** (1.869e-03)
Observations	184,898	163,088	21,810
R ²	0.778	0.809	0.404
Adjusted R ²	0.769	0.802	0.383
Note:	*p<0.1; **p<0.05; ***p<0.01		

New text “Final results are robust to using a negative binomial at this stage (Table S7).” Line 336

2. This brings me to an important point which I did not bring up on prior review, but I have just noticed (and I apologise I did not catch this on the first version). The authors are using regression methods to estimate Embeta_t, yet it appears they just use the central value of that quantity in the climate regression in Equation 6. I do not see a discussion in their methodology how uncertainties in the estimate of Embeta_t are incorporated into the analysis (or even if this is considered at all). Lewis and Linzer (2005) discuss the heteroskedasticity problems that arise with regressing a dependent variable that itself is an

estimate: <https://pdfs.semanticscholar.org/0575/172b931858d6a713f8f10d41ae8ad9104edb.pdf>

The problem is that not taking into account the uncertainty on Embeta_t can lead to over-estimation of the power of the analysis in detecting climate-dependent effects. An iterative weighted linear regression method such as the one Lewis and Linzer suggest in Section 4.2 of their paper should be used to get unbiased estimates of the confidence intervals on the regression parameter estimates.

It is worth noting here that if the uncertainty on Embeta_t is sigma, the uncertainty on log(Embeta_t) is w=sigma/Embeta_t, and 1/w should be the weight used in the log link linear regression.

We thank the reviewer for raising this important issue and apologize for neglecting to include information on our standard error treatment in the methods section. Our method uses clustered standard errors to account for heteroskedasticity.

As indicated in the suggested reference (and elsewhere), the concern with using an estimated dependent variable (EDV) is that the sampling uncertainty is not constant across observations, which might make the regression errors heteroskedastic [Lewis and Linzer 2005]. In our case, the sampling error is introduced in the estimation of Sbar which occurs at the location level. Accordingly, the potential emergent heteroskedasticity is controlled for with the use of clustered standard errors at the location scale, which allow for arbitrary dependence between observations within a cluster.

To ensure that this is, indeed, the case, we fitted the model using White’s standard errors (suggested by Lewis and Linzer as generally reliable), weighted least squares (Section 4.2 of Lewis and Linzer) and clustered standard errors (used in the main result). The table comparing these results is in the Supplementary Materials Table S4 (and reproduced below). The weighting approach tends to weight the parameter estimates slightly towards our US observations which are less noisy (due to more stringent national surveillance procedures) and therefore have a lower uncertainty in the EDV. However, since the qualitative conclusions are unchanged, and because we aim to provide a balanced representation of climate effects across these two locations, we retain our clustered results in our main regression.

	Dependent variable:		
	Log Transmission		
	(White)	(Weighted)	(Clustered)
1/Specific Humidity	2.415e-04*** (3.297e-05)	3.070e-04*** (2.534e-05)	2.415e-04 *** (2.665e-05)
Precipitation	1.853e-03*** (3.950e-04)	1.608e-03*** (4.811e-04)	1.853e-03*** (5.095e-04)
Observations	119,802	119,802	119,802
R ²	0.825	0.811	0.825
Adjusted R ²	0.818	0.804	0.818

Note: *p<0.1; **p<0.05; ***p<0.01

Table 1: A comparison of the main results using different standard error corrections.

New text: “Standard errors are clustered at the location level, though White’s standard errors give similar results (Table S4).” Line 367,

3. In my original review, I expressed concern that the authors were removing data where the R² was less than 0.5. As I stated in the original review, this likely biases the analysis results by over-inflating the estimated power. The authors state in their replies they have switched to a population cut-off instead, but the text of the revised version indicates they are still making this R² selection.

We apologise for not providing a clear response in the previous review. The results for the population cutoff are shown in the supplement. Using the population cutoff does not change our main results.

The aim of the first stage of our modeling process is to provide the best estimates of empirical beta. Removing locations with a poor TSIR fit improves the accuracy our empirical beta estimates. This does not bias our estimated effect of climate on transmission as the results are independent of climate drivers at this stage.

New text “Our results are robust to using the full dataset and a population-based cut off (Table S5, S6).” Line 349.

Reviewers' Comments:

Reviewer #2:

Remarks to the Author:

I am satisfied with the manuscript changes and have no further comments.

REVIEWERS' COMMENTS:

Reviewer #2 (Remarks to the Author):

I am satisfied with the manuscript changes and have no further comments.

We thank the reviewer for their helpful comments.